# Complications for a Hoyeraal–Hreidarsson Syndrome Patient with a Germline *DKC1* A353V Variant Undergoing Unrelated Peripheral Blood Stem Cell Transplantation

**DOI:** 10.3390/ijms20133261

**Published:** 2019-07-02

**Authors:** Rong-Long Chen, Kuanyin K Lin, Liuh-Yow Chen

**Affiliations:** 1Department of Pediatric Hematology and Oncology, Koo Foundation Sun Yat-Sen Cancer Center, Taipei 11259, Taiwan; 2Institute of Molecular Biology, Academia Sinica, Taipei 11529, Taiwan

**Keywords:** dyskeratosis congenita, Hoyeraal–Hreidarsson syndrome, portal hypertension, pulmonary arteriovenous shunts, reduced intensity conditioning, telomeropathy, unrelated peripheral blood stem cell transplantation, vascular aging

## Abstract

Hoyeraal–Hreidarsson syndrome (HHS), caused by several different germline mutations resulting in severe telomeropathy, presents with early-onset growth anomalies and neurologic/developmental disorders including characteristic cerebellar hypoplasia. Early mortalities may arise from immunodeficiency and bone marrow failure if not successfully salvaged by allogeneic hematopoietic stem cell transplantation (HSCT). Few reports have characterized the persistent somatic progression of HHS after successful HSCT. We present an HHS patient with an X-linked recessive *DKC1* c.1058C > T; Ala353Val mutation who successfully underwent unrelated HSCT at 5 years of age. After months of early infections and organ toxicities immediately post-transplant, he had more than two years of excellent quality of life with correction of bone marrow failure and immunodeficiency. However, episodic massive variceal bleeding and progressive respiratory insufficiency, which were secondary to non-cirrhotic portal hypertension and pulmonary arteriovenous shunts, respectively, developed over 2 years after HSCT and resulted in his death from respiratory failure 4 years after HSCT. This outcome suggests that while HSCT can correct bone marrow failure and immunodeficiency, it may fail to prevent or even aggravate other fatal processes, such as portal hypertension and pulmonary arteriovenous shunting.

## 1. Introduction

Hoyeraal–Hreidarsson syndrome (HHS, OMIM 500545)—a severe variant of dyskeratosis congenita (DC, OMIM 305000) involving very short leukocyte telomere length—is characterized by intrauterine growth retardation and cerebellar hypoplasia, in addition to the classical mucocutaneous DC triad (nail dystrophy, skin hyperpigmentation, oral leukoplakia), with early life-threatening presentations related to immunodeficiency and bone marrow failure (BMF) [1]. Germline variants resulting from different gene mutations—namely *DKC1*, *TINF2*, *TERC*, *TERT*, *NOP10*, *NHP2*, *CTC1*, *WRAP53*, *ACD*, *RTEL1*, and *PARN*—have been implicated in HHS and other DC variants. These variants have been documented as mainly perturbing telomeric functions, with the most severe disease forms presenting significantly shorter telomeres than milder forms [2]. Among affected gene variants, X-linked recessive male patients with the *DKC1* c.1058C > T; Ala353Val missense mutation present with a diverse clinical phenotype, ranging from late-onset DC without BMF to lethal HHS [2]. Given the high rate of early mortality due to BMF and/or immunodeficiency in HHS patients, there are very few reports in the literature on HHS patients with variable genetic variants that have received hematopoietic stem cell transplantation (HSCT) [3,4,5,6,7]. Herein, we report our experience of treating an HHS patient with a characterized *DKC1* mutation, who successfully recovered from early life-threatening complications after receiving a reduced intensity conditioning (RIC) preparation, followed by unrelated peripheral blood stem cell transplantation (PBSCT). However, regrettably, vascular aging events continued to evolve and resulted in mortality four years after HSCT.

## 2. Case Reports

The patient, a boy, suffered oligohydramnios during pregnancy and was born prematurely in December 2009, with a gestational age of 33 weeks and a birth body weight of 1318 g. Growth and developmental delay requiring rehabilitation were noted from early infancy. In addition, progressive skin hyperpigmentation, nail dysplasia, and leukoplakia of the tongue were also noted. In January 2011, aged 1 year, he was first noted to have cytopenia (leukocytes 5.22 × 10^9^/L, hemoglobin 8.1 mg/dL, platelet count 18 × 10^9^/L). He was treated with intravenous immunoglobulins and prednisolone, but the pancytopenia worsened. He required increasingly frequent transfusions and treatment for infections and, by late 2014, he had a neutrophil count of around 0.5 × 10^9^/L, hemoglobin ranging from 6 to 7 mg/dL, and a platelet count of less than 10 × 10^9^/L.

Severe aplastic anemia was confirmed by bone marrow studies in November 2014. In addition, he presented with severe immunodeficiency; very low CD16+56+ natural killer cell (0.013, reference 0.16–0.57 × 10^9^/L) and CD19+ B cell (0.001, reference 0.43–1.27 × 10^9^/L) counts were recorded. Moderate T-lymphopenia was also diagnosed; cell counts of CD3+ T cells = 0.415 (reference 1.58–3.71) × 10^9^/L, CD4+ T cells = 0.253 (reference 0.87–2.14) × 10^9^/L, and CD8+ T cells = 0.138 (reference 0.47–1.11) × 10^9^/L. Levels of immunoglobulins G, A, and M were within normal ranges for his age (data not shown). HHS arising from recurrent X-linked recessive *DKC1* c.1058C > T; Ala353Val mutation with extremely short telomeres was diagnosed (Figure 1). Phenotypes of this mutant variant include intrauterine growth retardation, severe lymphopenia, cerebellar hypoplasia (Figure 2), and the characteristic DC mucocutaneous triad of nail dystrophy, skin hyperpigmentation, and oral leukoplakia (Figure 3). He also had non-cirrhotic non-icteric hepatic biochemical abnormalities, but without evidence of pulmonary fibrosis or other lung pathologies.

As HLA-identical sibling donors were unavailable, a decision was made to pursue unrelated HSCT as salvage therapy. The conditioning consisted of alemtuzumab (a total of 1 mg/kg given from day −10 to day −6) and fludarabine (30 mg/m^2^ daily from day −8 to day −3), kindly provided by Boston Children’s Hospital [8]. For prophylaxis against graft-versus-host disease (GVHD), the patient received cyclosporine and mycophenolate mofetil. On 10 March 2015 (day 0), he received PBSCs from an unrelated donor comprising 45 × 10^8^ total nucleated cells/kg and 23 × 10^6^ CD34+ cells/kg. The patient and the donor were HLA-8/8 matched (A 1101/2402, B 4001/4601, Cw 0102/0702, DR 0809/0901) and ABO-matched (B to B). Neutrophil engraftment was documented on day +11. The patient did not require any transfusion after day +18. Chimerism analyses were performed from bone marrow (nine months post-transplantation) and from peripheral blood (repeatedly from day +27), all of which showed 100% donor chimerism.

The initial course was complicated by an episode of Hickman catheter-related *Kocuria rosea* sepsis and stage 2 skin acute GVHD. Concurrently with the development of skin GVHD, phimosis progressed with the appearance of an erythematous and bullous-like foreskin lesion. The lesion partially regressed following treatment with oral valacyclovir and prednisolone. In addition, diarrhea arose, with stool content changing from loose to watery then bloody, and was associated with cramping abdominal pain and intermittent fever. He required a long admission from day +51 to day +167, with complications including cytomegalovirus (CMV) reactivation, severe enterocolitis, buccal mucositis/cellulitis, and parotitis. He required parenteral nutritional support for nearly two months and developed Wernicke encephalopathy during this period that was responsive to thiamine treatment. Finally, these complications were alleviated and he was discharged under on-going prednisolone and cyclosporine treatment. 

However, he still experienced repeated CMV reactivation, episodes of diarrhea with or without fever, *Escherichia coli* urinary tract infection, *Clostridium difficile*-associated diarrhea, *Salmonella* group E enteric fever, and influenza during the first year post-transplantation. He also underwent circumcision because of a difficulty urinating, which was secondary to progressive obstructive phimosis at nine months post-transplantation. Steroid and cyclosporine treatments were discontinued 9 months and 1 year post-transplantation, respectively. Recovery of bone marrow hematopoiesis was documented (Figure 4). Moreover, the frequency of serious infections markedly decreased over the course of the second year post-transplantation, so the boy returned to school.

At the early post-transplantation stage, the patient’s hepatic abnormalities remained stationary, with mildly elevated transaminase levels, though hepatic ultrasonography revealed heterogeneous echotexture with multiple tiny hypoechoic nodules. He gradually gained body weight purely by oral intake from 12.4 kg before HSCT to 15.9 kg in November 2016. He had become transfusion-independent by 21 November 2016, with a neutrophil count of 2.69 × 10^9^/L, hemoglobin of 10.6 mg/dL, and a platelet count of 128 × 10^9^/L. A lymphocyte subset analysis in November 2016 revealed normal counts of CD16+56+ natural killer cells (0.184, reference 0.12–0.48 × 10^9^/L) and CD19+ B cells (0.390, reference 0.28–0.64 × 10^9^/L). The extent of T-lymphopenia had also decreased with CD3+ T, CD4+ T, and CD8+ T cell counts of 0.803 (reference 1.24–1.21), 0.272 (0.65–1.52), and 0.392 (0.37–0.95) × 10^9^/L, respectively.

However, massive upper gastrointestinal bleedings happened after respiratory tract infections in late April 2017 when portal hypertension complicated with esophago-gastric varices was diagnosed by emergency endoscopy and computerized tomography (Figure 5). In addition, he was noted to have persistent hypoxemia, with the arterial partial pressure of oxygen always less than 50 mmHg after treatment for a documented influenza episode in May 2017. The patient had very severe hepatopulmonary syndrome, with a calculated alveolar-arterial oxygen gradient of 70 mmHg at sitting and 75 mmHg at supine positions, respectively. Pulmonary arteriovenous shunting was documented by a lung perfusion scan in August 2017 (Figure 6). Only palliative management was considered thereafter. The patient was still able to enjoy school and home life under regular oxygen supplementation, although intermittent admissions were required to manage severe upper gastrointestinal bleeding or aggravated respiratory symptoms in his final days. Ultimately, he passed away in February 2019, primarily due to respiratory compromise.

## 3. Discussion

We have characterized herein the lifelong events of an HHS patient with *DKC1* A353V and extremely short telomeres. Despite the presence of perinatal intrauterine growth restriction and developmental delay, HHS was not diagnosed until the patient suffered from complications related to rapidly progressive BMF and immunodeficiency when the characteristic mucocutaneous DC triad and cerebral hypoplasia were noted.

HSCT is the first consideration for treatment when DC-associated BMF develops and a matched-sibling donor is available [9,10]. Until recently, HSCT for DC has been associated with an inferior outcome, mainly owing to early and late post-transplantation complications including graft failure, GVHD, sepsis, and, in particular, the increased propensity to develop pulmonary failure [11]. The largest cohort (*n* = 94) of non-myeloablative HSCT for DC still showed the crude mortality rate of 41%, with about one-third of deaths resulting from late complications such as organ damage and secondary malignancies [12]. For unrelated donor HSCT in DC, the risks are even higher, though RIC preparative regimens have been suggested as being more successful [8,13].

Instances of HSCT treatment for genotype-determined HHS are rarely reported. One of two sibling HHS with homozygous *TERT* mutation had early engraftment after HLA-9/10 unrelated bone marrow transplantations. However, another died one month after post-matched unrelated cord blood transplantation (CBT) [5]. Two HHS patients with the *RTEL* mutation were successfully rescued: one by haploidentical PBSCT and the other by unrelated CBT [6,7]. Another HHS patient with the *DKC1* mutation had delayed marrow reconstitution after an unrelated bone marrow transplant [3]. In contrast, our HHS patient (with the *DKC1* mutation) had prompt and complete donor engraftment, as well as steady immune recovery, after unrelated PBSCT following a novel radiation/alkylator-free RIC regimen [8]. Our patient gradually attained abilities to overcome mild and brief GVHD/infections.

The previously reported HHS patient with the *DKC1* mutation developed severe gastrointestinal problems that required chronic steroid therapy, long-term total parenteral nutrition, and jejunostomy feeding following an unrelated bone marrow transplant [3]. Another HHS patient with the *RTEL* mutation also required total parenteral nutrition at home for 2 years after CBT [7]. Protracted enterocolitis was also the most troublesome early post-transplantation complication in our patient, which has caused significant morbidities elsewhere [14]. We emphasize that the increased propensity of HHS patients to develop telomere-mediated intestinal disease (as reviewed by Jonassaint et al. [14]) should be highlighted as an early and life-threatening post-transplantation complication, even after RIC preparation.

The early success of HSCT in our patient failed to prevent (and may have even aggravated) as yet uncharacterized vascular aging processes that particularly affected the liver and lungs, causing severe non-cirrhotic portal hypertension and pulmonary arteriovenous shunting that resulted in death four years after HSCT. This scenario contrasts with those of germline telomerase mutations that predispose cirrhosis formation and familial idiopathic pulmonary fibrosis but, in general, do not result in the expression of classical DC phenotypes [15,16,17]. The vascular changes in our patient are similar to the setting of chronic liver diseases leading to hepatopulmonary syndrome, in which dilatation of pulmonary pre-capillary and capillary vessels is the striking pathological feature, and with some patients exhibiting a few pleural/pulmonary shunts as well as portopulmonary venous anastomoses [18]. Recently, pulmonary arteriovenous malformations, with or without hepatopulmonary syndrome, have been characterized as a pulmonary phenotype of DC, with 77% of such patients undergoing HSCT [19]. Among the 13 such cases reported, germline mutations were documented in twelve including *TINF2* (six), *TERT* (two), *RTEL1* (two), *PARN* (one), and *DKC1* T408I (one). We now add a case of a germline *DKC1* A353V HHS patient having pulmonary arteriovenous malformations with hepatopulmonary syndrome 2 years after HSCT.

In conclusion, a boy with characteristics of HHS caused by an X-linked recessive *DKC1* mutation developed life-threatening BMF and severe lymphopenia. These complications were successfully ameliorated over time through unrelated PBSCT using a novel radiation/alkylator-free RIC regimen with manageable early complications. He was subsequently able to return to school, but ultimately developed severe portal hypertension and fatal hepatopulmonary syndrome.

## 4. Materials and Methods

### 4.1. Whole Exome Sequencing (WES)

To identify genetic variants in the patient, genomic DNA from leukocytes of the subject was subjected to WES. The sequence data was aligned to a reference genome (hg19) to identify genetic variants. Analysis of the variants in genes associated with dyskeratosis congenita led to the identification of the *DKC1* c.1058C>T mutation. Sanger sequencing was carried out to verify the *DKC1* mutation. A fragment of DNA containing the *DKC1* mutation site was amplified by PCR using oligonucleotide primers 5′-tgagctgcaagcctgttatg-3′ and 5′-caaatccccctctgtgagaa-3′, which were also used for direct sequencing. Our results confirmed the *DKC1* mutation in the patient and revealed the mother to be a carrier with wild-type and mutant DKC1 alleles. Written informed consent was obtained from all the participants for public release of the survey results.

### 4.2. Telomere Length Analysis

Leukocyte telomere length was analyzed by telomere restriction fragment assay, as previously described [20]. Briefly, leukocyte genomic DNA was digested by Rsa I and Hinf I restriction enzymes and resolved by agarose gel electrophoresis. Telomeric DNA was detected by in gel hybridization using a [^32^P]-labeled telomeric probe.

### 4.3. Tecnetium-99m MAA Lung and Brain Scanning

A technetium-99m-labeled macroaggregated albumin lung perfusion scan was performed according to a previous report [21]. The extrapulmonary shunt fraction, assuming that 13% of the cardiac output is delivered to the brain, was calculated using the geometric mean of technetium (GMT) counts around the brain and lung according to the following formula: (GMTbrain/0.13)/ ((GMTbrain/0.13) + GMTlung).

## Figures and Tables

**Figure 1 ijms-20-03261-f001:**
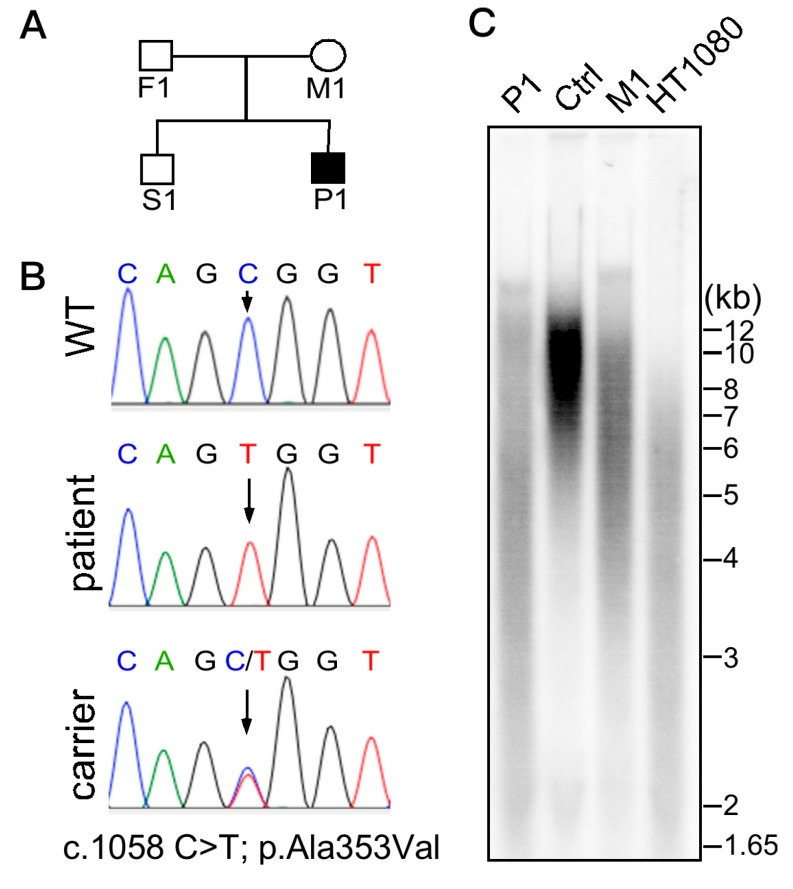
(**A**) Pedigree of the affected family. Solid symbol represents the affected patient and open symbols represent unaffected relatives. Squares indicate males and circle indicates a female subject. (**B**) Sanger DNA sequencing of *DKC1* exon 11 on chromosome X from peripheral blood cells taken from subjects of the studied pedigree. Wild-type (S1), patient (P1), carrier (M1). (**C**) Telomere length analysis (by terminal restriction fragment assay) of DNA from leukocytes of the patient (P1), an age-matched control (Ctrl), a carrier (M1), and from HT1080 fibrosarcoma cancer cells. The patient (P1) had exceedingly shortened and heterogeneous telomeres compared to the control (Ctrl).

**Figure 2 ijms-20-03261-f002:**
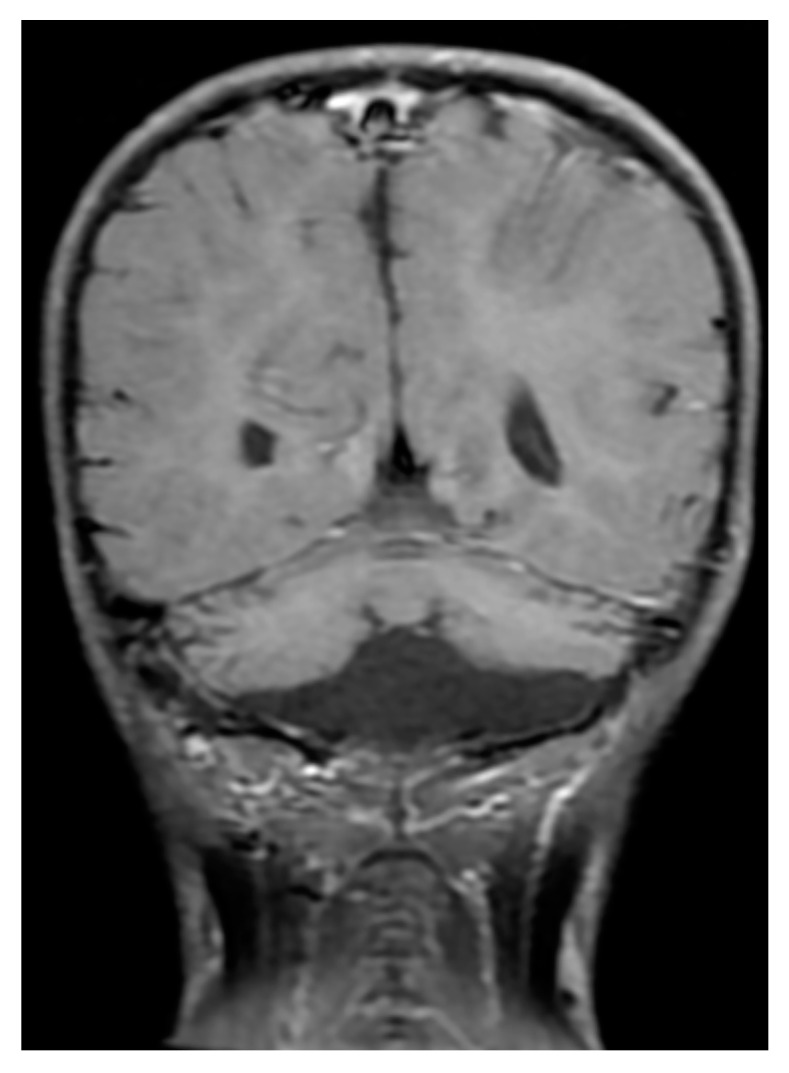
Brain magnetic resonance imaging of the patient in December 2014 showing characteristic cerebellar hypoplasia.

**Figure 3 ijms-20-03261-f003:**
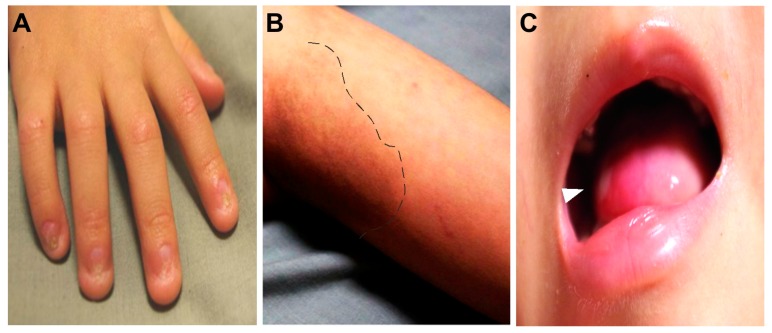
Pre-transplant photographs of the patient showing nail dysplasia (**A**), abnormal skin pigmentation (**B**), and tongue leukoplakia (white arrowhead) (**C**).

**Figure 4 ijms-20-03261-f004:**
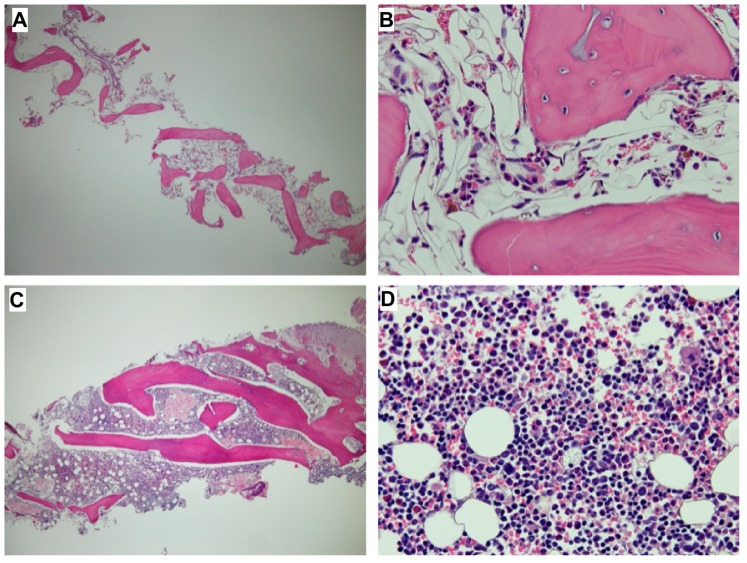
H&E-stained micrographs (40× **A**,**C**; 400× **B**,**D**) showing the severe hypocellularity with scanty hematopoiesis of bone marrow obtained pre-hematopoietic stem cell transplantation (HSCT) in February 2015 (**A**,**B**) compared to much more cellular material with abundant hematopoiesis obtained in December 2015 (i.e., nine months post-HSCT) (**C**,**D**).

**Figure 5 ijms-20-03261-f005:**
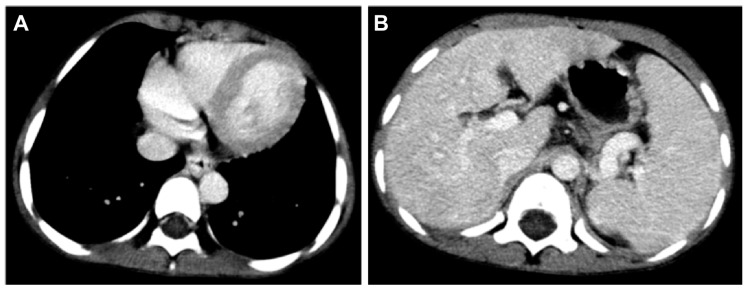
Enhanced computerized tomography scans showing esophageal (**A**) and gastric (**B**) varices.

**Figure 6 ijms-20-03261-f006:**
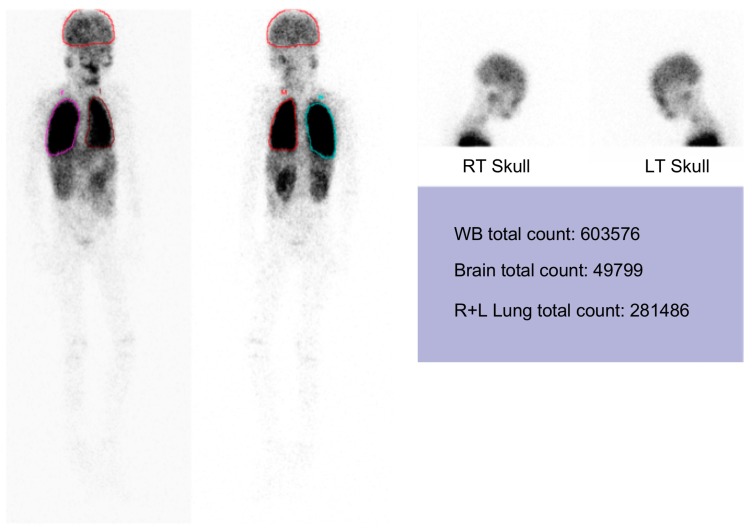
Technetium-99m-labeled macroaggregated albumin (Tc-99m MAA) dynamic perfusion imaging and total-body scans showing abnormal increased uptake in the brain immediately after Tc-99m MAA injection, with calculated intrapulmonary right-to-left shunts of 57.6%. LT, left; RT, right; R+L, right plus left; WB, whole body.

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
