# Peer review of "Complications for a Hoyeraal–Hreidarsson Syndrome Patient with a Germline *DKC1* A353V Variant Undergoing Unrelated Peripheral Blood Stem Cell Transplantation"

_ijms, 2019, doi:10.3390/ijms20133261_

Round 1
Reviewer 1 Report
In this study, the authors reported an Hoyeraal–Hreidarsson syndrome (HHS) patient with X-linked recessive DKC1 c.1058C>T; Ala353Val mutation, who successfully underwent unrelated allogeneic hematopoietic stem cell transplantation (allo-HSCT). However, he became the portal hypertension by pulmonary arteriovenous shunts after allo-HSCT in two years and died of the next two years. There are very few reports in the literature on HHS patients that have received allo-HSCT. Therefore, I think that the purpose of this article is interesting, however several points need to be clarified and certain statements are required for further justification.
Minor
1. Telomere length of a patient is unclear. The authors should show more clear result.
2. The authors should show a conditioning regimen of allo-HSCT in detail.
3. Fioredda F et al. collected data on the outcome of allo-HSCT in the largest cohort of Dyskeratosis congenita (n = 94) patients (Br J Haematol. 2018 Oct;183(1):110-118). This article is the latest article of the allograft for DKC. The authors quote this article and should discuss it.
Author Response
Point-by-point responses
We wish to thank the referee for very careful evaluation of our manuscript and valuable suggestions that have helped us to improve this manuscript. Below,
I provide a point-by-point explanation of how we have addressed the comments.
Reviewer #1
Comments and Suggestions for Authors
In this study, the authors reported an Hoyeraal–Hreidarsson syndrome (HHS) patient with X-linked recessive DKC1 c.1058C>T; Ala353Val mutation, who successfully underwent unrelated allogeneic hematopoietic stem cell transplantation (allo-HSCT). However, he became the portal hypertension by pulmonary arteriovenous shunts after allo-HSCT in two years and died of the next two years. There are very few reports in the literature on HHS patients that have received allo-HSCT. Therefore, I think that the purpose of this article is interesting, however several points need to be clarified and certain statements are required for further justification.
Response: Thank you for the comments and suggestions.
Minor
1. Telomere length of a patient is unclear. The authors should show more clear result.
Response: We have elaborated more on the description about telomere length … The patient (P1) had exceeding shortened and heterogeneous telomeres compared to the control (Ctrl)..in Page 4 Figure 1C legend, Line 136-137.
2. The authors should show a conditioning regimen of allo-HSCT in detail.
Response: We have included more detailed information of the conditioning regimen in Page 2, Line 77-78: The conditioning consisted of alemtuzumab (a total of 1 mg/kg given from day -10 to day -6) and fludarabine (30 mg/m2 daily from day -8 to day -3), kindly provided by Boston Children’s Hospital [8].
3. Fioredda F et al. collected data on the outcome of allo-HSCT in the largest cohort of Dyskeratosis congenita (n = 94) patients (Br J Haematol. 2018 Oct;183(1):110-118). This article is the latest article of the allograft for DKC. The authors quote this article and should discuss it.
Response: We have added this article as new reference 12 and discussed in Page 7, Line 169-171. A largest cohort (n = 94) of non-myeloablative HSCT for DC still showed the crude mortality rate of 41% with about one third of deaths resulting from late complications such as organ damage and secondary malignancies [12].
Reviewer 2 Report
In this manuscript, the authors describe a case report of Hoyeraal-Hreidarsson syndrom due to a mutation in DKC1. The report is well done with precise clinical description of the case.
However, the chromosomal localization of DKC1 gene on X chromosome is not mentioned in the text. This must be corrected and discussed with respect of publications showing that biology disorders may be or may not be present in females with DKC1 mutation.
Author Response
Point-by-point responses
We wish to thank the referee for very careful evaluation of our manuscript and valuable suggestions that have helped us to improve this manuscript. Below,
I provide a point-by-point explanation of how we have addressed their comments.
Reviewer #2
Comments and Suggestions for Authors
In this manuscript, the authors describe a case report of Hoyeraal-Hreidarsson syndrom due to a mutation in DKC1. The report is well done with precise clinical description of the case.
Response: We appreciate your comments.
However, the chromosomal localization of DKC1 gene on X chromosome is not mentioned in the text. This must be corrected and discussed with respect of publications showing that biology disorders may be or may not be present in females with DKC1 mutation.
Response: This point is well taken. We have added …X-linked recessive male… in Page 1, Line 43 and …on chromosome X… in Page 4, Line 133 to highlight the chromosomal localization of DKC1 gene on X chromosome.